# Ovulation Enhances Intraperitoneal and Ovarian Seedings of High-Grade Serous Carcinoma Cells Originating from the Fallopian Tube: Confirmation in a Bursa-Free Mouse Xenograft Model

**DOI:** 10.3390/ijms23116211

**Published:** 2022-06-01

**Authors:** Che-Fang Hsu, Vaishnavi Seenan, Liang-Yuan Wang, Tang-Yuan Chu

**Affiliations:** 1Center for Prevention and Therapy of Gynecological Cancers, Department of Medical Research, Hualien Tzu Chi Hospital, Buddhist Tzu Chi Medical Foundation, Hualien 970, Taiwan; cfhsu@tzuchi.com.tw (C.-F.H.); 104727118@gms.tcu.edu.tw (V.S.); 106712141@gms.tcu.edu.tw (L.-Y.W.); 2Institute of Medical Sciences, Tzu Chi University, Hualien 970, Taiwan; 3Department of Molecular Biology and Human Genetics, Tzu Chi University, Hualien 970, Taiwan; 4Department of Obstetrics & Gynecology, Hualien Tzu Chi Hospital, Buddhist Tzu Chi Medical Foundation, Hualien 970, Taiwan; 5Department of Life Sciences, Tzu Chi University, Hualien 970, Taiwan

**Keywords:** ovulation, follicular fluid, high-grade serous carcinoma, fallopian tube, intraperitoneal seeding, humanized mouse models

## Abstract

**Background:** Recently, new paradigms for the etiology and origin of ovarian high-grade serous carcinoma (HGSC) have emerged. The carcinogens released during ovulation transform fallopian tube epithelial cells, exfoliating and metastasizing to the peritoneal organs, including the ovaries. Solid in vivo evidence of the paradigms in a mouse model is urgently needed but is hampered by the differing tubo-ovarian structures. In mice, there is a bursa structure surrounding the distal oviduct and ovary. This, on one hand, prevents the direct influence of ovulatory follicular fluid (FF) on the exfoliated tumor cells. On the other hand, it hinders the seeding of exfoliated tumor cells into the ovary. **Methods:** In this study, we created a bursa-free mouse xenograft model to examine the effect of superovulation on peritoneal and ovarian metastases of transformed human tubal epithelial cells after intraperitoneal injection in NSG mice. **Results:** The bursa-free mouse model showed a better effect of ovulation on peritoneal metastasis. In this model, superovulation increased the number of transformed human tubal epithelial cell seedlings after intraperitoneal injection. Compared to the bursa-intact state, bursa-free ovaries were more vulnerable to external tumor seeding in either normal ovulation or superovulation state. **Conclusions:** This study provides the first in vivo evidence that intraperitoneal spreading of tubal HGSC cells is enhanced by ovulation. This study also demonstrated a mouse model for studying ovary-peritoneum interaction in cancer development.

## 1. Introduction

High-grade serous carcinoma (HGSC) is the most common type of epithelial ovarian cancer (EOC) and is a major killer. Nearly 80% of cases are diagnosed after intraperitoneal spread, leading to a poor prognosis. The problem has not improved over the past several decades, mainly because of the illusive etiological cause, mechanisms of malignant transformation, and even the tissue of origin. The puzzle of ovarian HGSC origin has been largely resolved. Multiple levels of evidence have indicated that secretory cells in the fallopian tube fimbrial epithelium (FTE) are the main origin of HGSC in the ovary, fallopian tube, and peritoneum [1,2,3].

Results from multiple epidemiological studies are consistent with the theory of incessant ovulation as the cause of EOC [4,5]. Factors that reduce the number of ovulation cycles, such as oral contraceptive use, full-term pregnancy, and breastfeeding, all showed a protective effect on HGSC [6]. Thus, a woman’s number of ovulation cycles is associated with an increased EOC risk [7].

The mechanisms of the ovulation-induced FTE transformation are becoming increasingly clear. Ovulatory follicular fluid (FF) contains high levels of various carcinogens, such as reactive oxygen species (ROS), insulin-like growth factor (IGF) axis proteins, and coagulation factor-hepatocyte growth factor (HGF) activation cascade proteinases, which confer a range of cell transformation phenotypes, including DNA double-strand breaks [8,9], stemness activation, clonal expansion [10], anchorage-independent growth, and xenograft tumorigenesis [10,11]. The transformation effect of ovulatory FF is not limited to the FTE transformation. It also confers metastatic seeding of transformed FTE cells by promoting anoikis resistance, peritoneal attachment, proliferation, migration, and invasion [12]. These results suggest that ovulation not only promotes the malignant transformation of FTE but also the peritoneal spreading of tubal intraepithelial carcinoma (STIC), the immediate precursor of HGSC [13,14].

In all the previous studies, the pro-metastasis activities of ovulation were demonstrated by human FF exposure in vitro or ex vivo [8,10,11] or by co-injection of FF, transforming human FTE cells into the peritoneal cavity of immune-compromised mice [9,12]. An in vivo demonstration of the effect of ovulation on HGSC development is lacking. Before experimental proof is feasible, an anatomic barrier in the murine reproductive system must be overcome. In rodents, a bursa structure envelops the ovaries and distal oviduct [15] and interferes with the direct communication of FF with the peritoneal cavity after ovulation [16]. The presence of ovarian bursa in mouse not only prevents the direct release of ovulatory FF into the peritoneum but may also hinder the seeding of exfoliated STIC cells into the ovary.

In the present study, we established a corresponding model to overcome the bursa barrier and demonstrated the metastasis-enhancing effect of ovulation in vivo for the first time.

## 2. Results

### 2.1. Establishment of a Bursa-Free Ovulation Model for i.p. Xenograft

Figure 1A shows the timeline of the surgical removal of the ovarian bursa and superovulation in the studied mice. To avoid surgical wound disturbance [17], an i.p. xenograft was performed ten days after the operation. In the superovulation group, FEXT2-LUC cells were injected 6 h after hCG administration or 6 h before the expected ovulation time. Intraperitoneal tumor growth was monitored weekly using an IVIS for 6 weeks. Mice were sacrificed in the seventh week. Representative images of bursa-intact and bursa-free ovaries are shown in Figure 1B.

### 2.2. Removal of Ovarian Bursa Increased Periovarian Tumor Seeding While Making No Difference of Peritoneal Seeding of i.p. Injected Transformed FTE Cells

As shown in Figure 2A and Table 1, bursa-free and bursa-intact mice did not show differences in intraperitoneal tumor luminescence signal throughout the follow-up period. There was no difference in the number or total weight of intraperitoneal tumors examined at the seventh week (data not shown). As expected, the removal of the ovarian bursa led to an increase in periovarian seeding of PAX8(+) tumors in both normal-ovulation and superovulation mice (*p* = 0.009 and *p* = 0.02, respectively, and *p*= 0.0006 as a pool) (Figure 2B,C, Table 1).

### 2.3. Superovulation Enhanced the Early Peritoneal Seeding and The Later Growth of Transformed FTE Cells, Which Was Revealed Only in the Bursa-Free Xenograft Model

On the third day after intraperitoneal injection, bursa-intact mice, with or without superovulation, did not show significant differences in the tumorigenesis signal. However, in the bursa-removed group, superovulation mice showed a significantly higher intraperitoneal tumor signal than normal ovulation mice (Figure 3A, Table 1), and the differences increased over time (Figure 3B). Seven weeks after i.p. injection, there was also a higher average tumor weight (8.9 folds, *p* = 0.049) and tumor number (2.0 folds, *p* = 0.053) in the superovulation group than in the normal ovulation group (Figure 4A). This was again observed only in bursa-removed mice.

### 2.4. Superovulation in the Bursa-Free Status Did Not Further Increase the Periovarian Tumor Seeding

Unexpectedly, under the bursa-free setup, there was no statistical difference in the periovarian seeding of xenograft tumors between the normal ovulation and superovulation groups (Figure 4B).

## 3. Discussion

There is a bursa structure in rodents enclosing the ovary and distal oviduct. It helps to confine the ovulated oocytes and facilitate the catch-up by the oviduct infundibulum. Although the small opening (of approximately 0.04 to 0.12 cm^2^ [15]) of the foramen of the ovarian bursa located at the proximal mesosalpinx allows limited drainage, the majority of ovulatory FF and oviduct fluid flowing into the uterine cavity instead of the peritoneal cavity after ovulation [15,18]. It is thus assumed that the bursa structure, on one hand, interferes with the release of FF-carcinogens into the peritoneum and, on the other hand, with the seeding of exfoliated cancer cells into the ovary. Indeed, the present study showed an increase in local ovarian seeding after bursa removal.

The presence of the bursa structure retards the release of pro-metastatic FF factors into the peritoneum, thereby reducing intraperitoneal seeding of intraperitoneally injected tumor cells. In the STIC-to-HGSC stage of cancer development, STIC cells frequently detach from the epithelium into the lumen of the fallopian tube. Detached cancer cells usually float in the peritoneal fluid as multicellular spheroids [19]. Due to their miniature size, how these exfoliated STIC cells metastasize intraperitoneally is largely unknown. We previously discovered that i.p. co-injection of human FF with HGSC cells or transforming FTE cells promotes intraperitoneal metastasis [12]. FF harbors a pro-metastasis activity, which favors anoikis resistance, anchorage-independent growth, peritoneal attachment, proliferation, and migration of HGSC cells and STIC-mimicking partially transformed FTE cells [12]. The present study provides the first in vivo evidence that endogenous ovulation promotes intraperitoneal metastasis of transformed FTE cells.

Ovulatory FF harbors various molecules essential for folliculogenesis, ovulation, and postovulatory tissue regeneration [20]. Many of these molecules play essential roles in oncogenesis and metastasis. These include cytokines (IL-1β, IL-6) [21,22], chemokines (SDF-1) [23,24], growth factors (IGF2, HGF, TGF-β1, PDGF, FGF, and VEGF) [25,26,27,28,29], soluble ECM proteins (collagen I, III, IV, fibronectin, and vitronectin) [30,31,32,33], and adhesion molecules (ICAM-1and VCAM-1) [34,35,36]. Moreover, ovulation is a consequence of acute inflammation after luteinization of the ovarian follicle, which releases rich ROS, inflammatory cytokines and cells into the peritoneal cavity. Pelvic inflammation is a known risk factor for ovarian cancer [37,38]. Thus, ovulation may contribute to an inflammatory tumor microenvironment that promotes cell proliferation, angiogenesis, and cancer metastasis [39].

In our bursa-free i.p. injection xenograft model, all seven tested mice grew tumors in the normal ovulation state. Superovulation, while also achieving a 100% tumorigenesis rate, resulted in a similar severity of local tumorigenesis as measured by the area of PAX8+ tumor lesions (Figure 2C). We suppose that under the oncogenic influence of normal ovulation, orthotopic tumorigenesis from ovarian seeding is already saturated, and no further increase can be achieved by superovulation.

The bursa-free mouse model, although essentially mimicking the ovulatory FF release in humans, is still limited by the fundamental anatomic and physiological differences between mice and humans. First, although we did not find a regeneration of the bursa up to seventh weeks after the removal, whether the bursa-free state will affect ovarian physiology, such as the efficiency of ovulation and the postovulatory wound healing, is not known. The second concern is whether the direct drainage of FF into the peritoneal cavity will change the constitution of peritoneal fluid or affect the peritoneal environment. Third, after mimicking bursa-free ovulation, a robust model simulating menstruation and its backflow is still lacking, although there is a way to induce decidualization and blood-shedding in rodent endometrium by hormone and sesame oil treatment [40]. Finally, the current in vivo model, although conveying the effect of ovulation on the peritoneal metastasis, is still not a spontaneous orthotopic tumorigenesis model of HGSC. A genetically engineered mouse model targeting the initiating driver mutations of *TP53* and *CCNE1* in the LGR-expressing cancer initiation cells in FTE will be an ideal model to test the effect of ovulation on the genesis and metastasis of HGSC.

In conclusion, the bursa-free mouse model better simulates human ovulation and facilitates the observation of the effect of ovulation on cancer metastasis. This study provides the first in vivo proof that superovulation enhances the peritoneal seeding of transformed FTE cells. A scenario of metastatic peritoneal seeding of STIC cells promoted by ovulation can thus be validated. The established bursa-free mouse model may provide an important tool for the study of various postovulatory impacts, such as ovulation-related wound healing and adhesions, as well as the development of endometriosis and cancer metastasis.

## 4. Material and Methods

### 4.1. Cell Culture

The HPV E6/E7/hTERT-immortalized human FTE cell line FE25 has been previously described [8]. FEXT2 was derived from a xenograft tumor of FE25 cells transformed by human FF after i.p. co-injection into NSG mice [41]. The FEXT2-LUC cell line was derived from FEXT2 cells by transduction with luciferase-expressing lentivirus pLAS3w.FLuc.Puro (National RNAi Core Facility of Academia Sinica, Taipei, Taiwan). FEXT2-LUC cells were maintained in MCDB105/M199 medium (1:1, Merck, NJ, USA) supplemented with 10% fetal bovine serum (FBS, Thermo Fisher Scientific, Waltham, MA, USA), 10 mg/mL puromycin (Invitrogen, San Diego, CA, USA), and penicillin/streptomycin (P/S, Corning Inc., Corning, NY, USA).

### 4.2. Bursa Removal Surgery and Xenograft Model

To analyze the abdominal dissemination of transformed human FTE cells, we performed bursectomy and i.p. xenografts of FEXT2-LUC cells in 8- to 12-week-old female NOD/Shi-scid/IL-2Rγ^null^ (NSG) mice (Jackson Laboratory, Bar Harbor, ME, USA). Briefly, after hair removal and disinfection, the uterine horn was exposed by a 1-cm incision at the dorsal skin. Under a four-fold dissecting surgical microscope, the fat behind the ovary was held with forceps, and the bursa was peeled off with forceps (Figure 1B). The surgical wound was closed with a skin nail, and the same procedures were performed on the other side. The entire procedure takes approximately 10–15 min. In the sham operation, the uterine horn was exposed using the same laparotomy procedure and was left in the air for the same duration.

Seven days after the operation, when the wounds were completely healed, mice were superovulated with 5 IU PMSG and 5 IU hCG (given 46 h apart) in a 4-day cycle for 6 weeks. Six hours after the first hCG injection, FEXT2 cells (1 × 10^4^) were mixed with 200 µL PBS and injected into the peritoneal cavity (as shown in Figure 1A). Based on the status of bursa and ovulation, mice were grouped into bursa-intact normal ovulation (BINO), or superovulation (BISO), and bursa-removed normal ovulation (BRNO) or superovulation (BRSO) groups, and followed weekly by an in vivo imaging system (IVIS^®^, PerkinElmer, Shelton, CT, USA). All mouse study procedures were performed according to protocols approved by the Institutional Animal Care and Use Committee of Tzu-Chi University (Approval ID: 108–25).

### 4.3. Scoring of the PAX8-Expressing Periovarian Metastasis Lesions

Immunohistochemistry (IHC) of the HGSC-specific marker PAX8 [42] was performed in tubo-ovarian tissue collected from mice 7 weeks after i.p. xenografts. For lesion size scoring, tubo-ovarian tissues of the BINO, BISO, BRNO, and BRSO mice were randomly sampled (*n* = 5, 7, 7, and 6, respectively) and subjected to paraffin embedding, followed by tissue section, and IHC staining using the UltraVision™ Quanto Detection System (Thermo Fisher). The slides were incubated overnight at 4 °C with an anti-PAX8 polyclonal rabbit primary antibody (ab53490, Abcam, Inc.) with 1:100 dilution and stained with DAB. The area of PAX8-expressing lesions was scored semiquantitatively under the 40× field of the microscope (Zeiss, Axio Vert A1, Oberkochen, Germany) by calculating the pixel^2 of the DAB(+) area using Image J software (Rasband, W.S., ImageJ, U.S. National Institutes of Health, Bethesda, MD, USA) [43].

### 4.4. Statistics

All data are presented as the mean ± standard error. Statistical analyses were performed using GraphPad Prism version 8.0 (GraphPad Software, La Jolla, CA, USA) and Microsoft Excel 2019. Detailed information on the statistical analysis is provided in the figure legends.

## Figures and Tables

**Figure 1 ijms-23-06211-f001:**
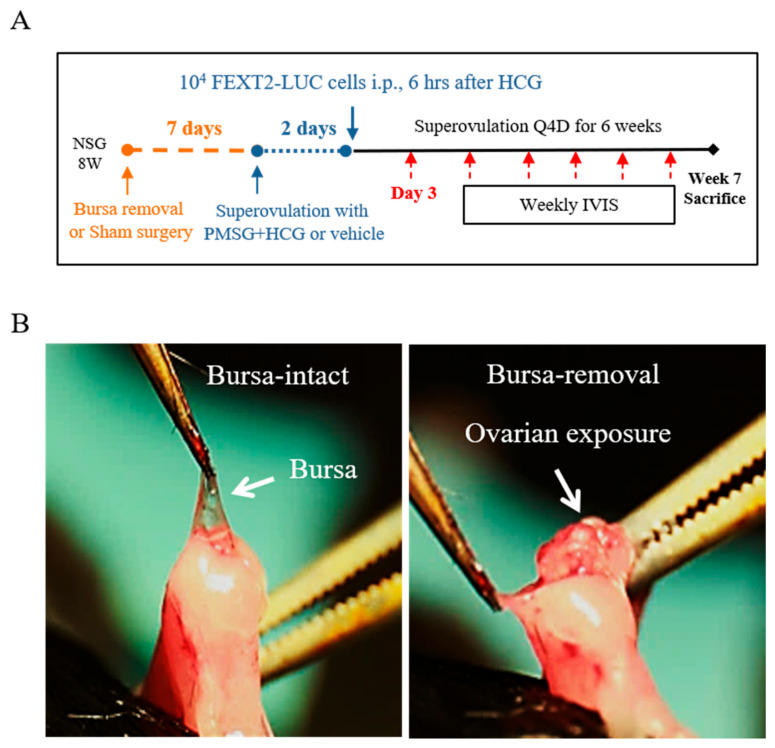
Schema of bursa removal, superovulation, and intraperitoneal injection in NSG mice. (**A**) Timeline of animal procedures. On the seventh day after the bursa removal or sham operation, mice were subjected to superovulation with PMSG/HCG or PBS as a control every 4 days until sacrifice. Two days later, 1 × 10^4^ FEXT2-LUC cells were injected i.p. Mice were followed weekly with IVIS. (**B**) Symbolic picture for bursa removal surgery.

**Figure 2 ijms-23-06211-f002:**
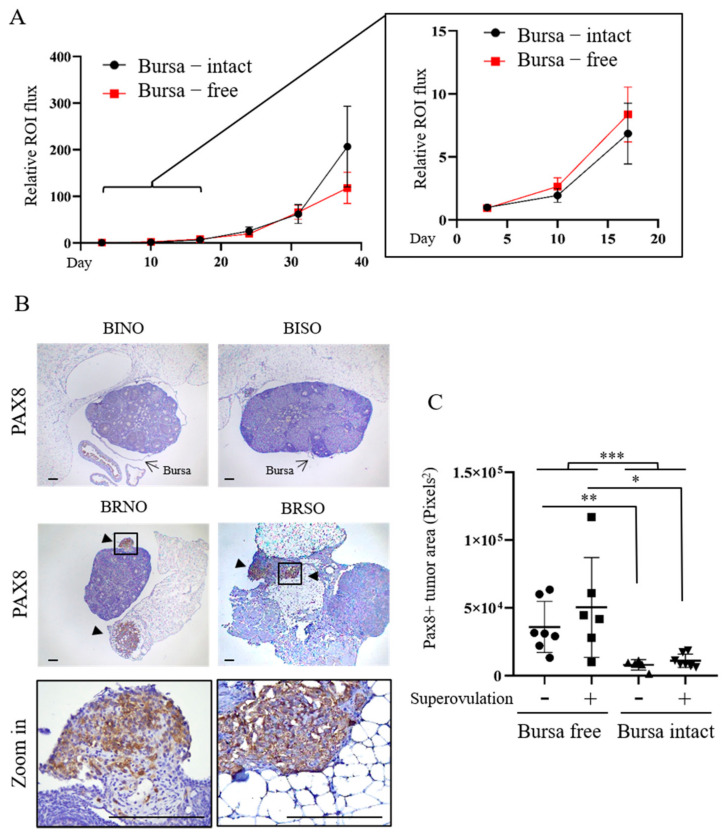
The bursa-free state disclosed the ovarian homing of transformed FTE cells from the peritoneal cavity. (**A**) Mean (±SEM) intraperitoneal IVIS flux in the region of interest (ROI) at different time points in 7 bursa-intact and 11 bursa−removed normal ovulation mice. (**B**) Examples of PAX8−expressing periovarian lesions examined in the seventh week. (**C**) Areas of PAX8(+) lesion in randomly sampled tubo−ovarian tissues from different mouse groups, as described in Materials and Methods. * *p* < 0.05, ** *p* < 0.01, *** *p* < 0.001 by two−sided, unpaired Student’s *t*-test.

**Figure 3 ijms-23-06211-f003:**
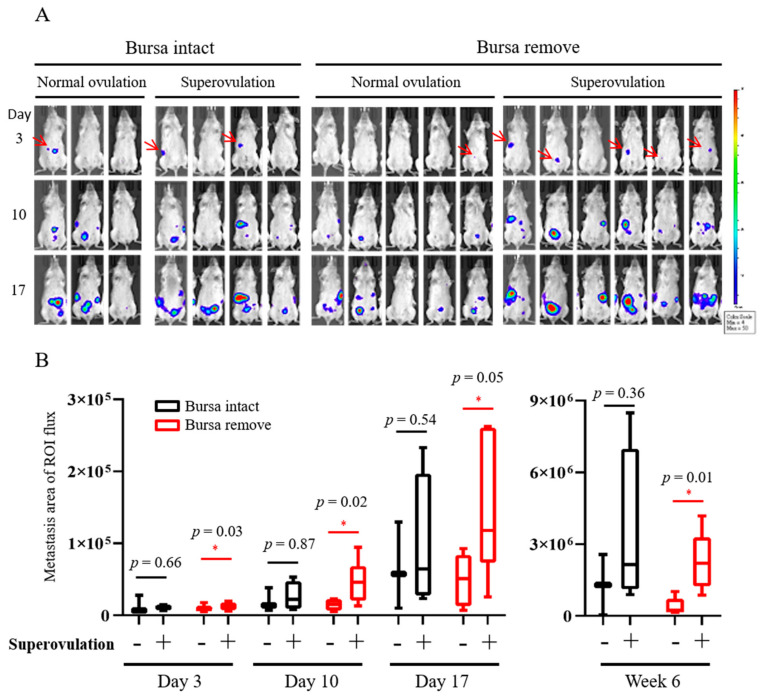
The bursa-free model better displays the effect of ovulation on peritoneal seedings of transformed FTE cells. (**A**) IVIS live images taken on the third, tenth, and seventeenth days after i.p. xenograft. Red arrows indicate the small lesion signal observed on the third day. (**B**) Mean IVIS signals of different mouse groups taken on days 3, 10, 17, and week 6. * *p* < 0.05, by two−sided, unpaired Student’s *t*-test.

**Figure 4 ijms-23-06211-f004:**
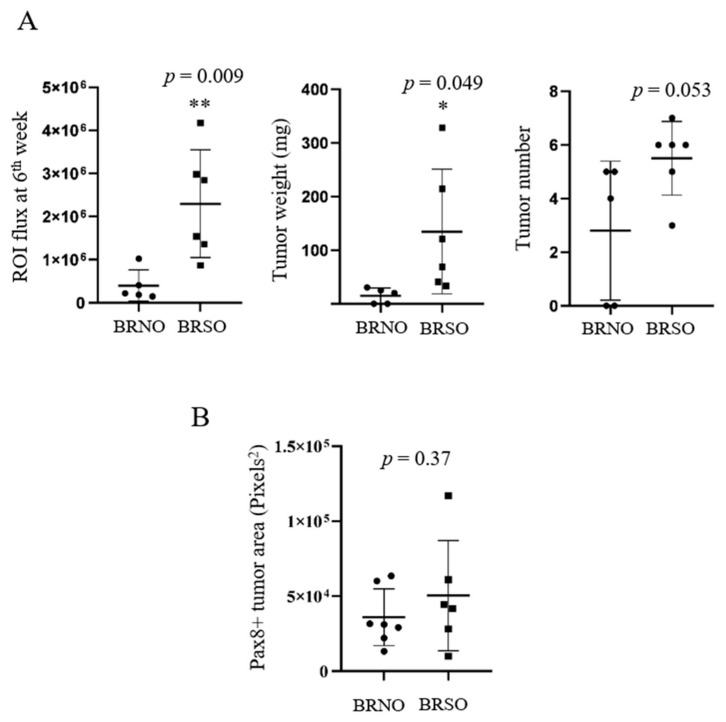
Compared to normal ovulation, superovulation resulted in an increase in peritoneal seedings but not local ovarian seeding of transformed FTE cells. (**A**) IVIS ROI flux signals, tumor weight, and tumor number measured in the sixth week after i.p. xenograft in different groups. BR: Bursa−removed, BI: Bursa-intact, NO: Normal ovulation, SO: Superovulation. * *p* < 0.05, ** *p* < 0.01, by two-sided, unpaired Student’s *t*-test. (**B**) PAX8-expressing tumor area in bursa-removed mice with or without superovulation.* *p* < 0.05, ** *p* < 0.01, by two−sided, unpaired Student’s *t*-test.

**Table 1 ijms-23-06211-t001:** Intraperitoneal tumor signal detected by IVIS at different time points in different mouse groups.

	Bursa-Intact						Bursa-Removed						Bursa-Removed vs. Bursa-Intact			
	Normal Ovulation (*n* = 3)		Super-Ovulation (*n* = 4)		SO vs. NO		Normal Ovulation (*n* = 5)		Super-Ovulation (*n* = 6)		SO vs. NO		Normal Ovulation		Super-Ovulation	
	Intraperitoneal ROI Flux		Intraperitoneal ROI Flux		Ratio	*p* Value	Intraperitoneal ROI Flux		Intraperitoneal ROI Flux		Ratio	*p* Value	Ratio	*p* Value	Ratio	*p* Value
Day 3	12,991	±9811	10,558	±3601	0.81	NS	6764	±2175	13,064	±4865	1.93	0.025	0.52	NS	1.24	NS
Day 10	32,636	±23,253	35,830	±23,584	1.10	NS	15,764	±13,659	55,280	±29,210	3.51	0.005	0.48	NS	1.54	NS
Day 17	92,660	±69,879	135,315	±93,062	1.46	0.08	52,280	±47,498	169,238	±100,547	3.24	0.006	0.56	NS	1.25	NS
Week 6	1,296,383	±1,268,437	3,418,550	±3,439,943	2.64	0.16	430,920	±362,705	2,297,667	±1,248,648	5.33	0.001	0.33	NS	0.67	NS

SO: super-ovulation, NO: normal ovulation, NS: non-significant.

## Data Availability

Not applicable.

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
