# Peer review of "Ovulation Enhances Intraperitoneal and Ovarian Seedings of High-Grade Serous Carcinoma Cells Originating from the Fallopian Tube: Confirmation in a Bursa-Free Mouse Xenograft Model"

_ijms, 2022, doi:10.3390/ijms23116211_

Round 1

Reviewer 1 Report

In this manuscript by Hsu et al, authors report a new model of bursa-free mouse xenograft to investigate intraperitoneal and ovarian seedings of high-grade serous carcinomas. This is an interesting study and a good model that better reflect human disease due to bursa removal. One key observation reported by the authors is that "superovulation enhances peritoneal seeding of transformed FTE cells."

This conclusion was not conclusively supported by the study results. Referencing to Figure 3B at Week 6, tumor burden in bursa-intact (black boxplot) mice have higher values than in bursa-free (red boxplot) mice.

Current study is limited to one cell line. Inclusion of another cell line would further strengthen the study.

Figure 2C. Results are not unexpected. Intact bursa will present seeding of transformed FTE cells to the ovary. Thus, Pax8+ cells in the tubo-ovarian samples in Bursa intact mice are considerably less than those in Bursa free mice.

If the key conclusion is that normal or superovulation releases factors that promote peritoneal seeding of HGSC cells, evidence provided are not convincing considering that bursa removed mice have lower tumor burden at week 6. 

Author Response

REVIEWER 1:

This conclusion was not conclusively supported by the study results. Referencing to Figure 3B at Week 6, tumor burden in bursa-intact (black boxplot) mice have higher values than in bursa-free (red boxplot) mice.

If the key conclusion is that normal or superovulation releases factors that promote peritoneal seeding of HGSC cells, evidence provided are not convincing considering that bursa removed mice have lower tumor burden at week 6. 

Response:
As indicated in the above original Figure 3B, shown by the arrows, the two mouse groups were under normal ovulation. It was not a super-ovulation vs. normal ovulation comparison. It was a comparison of bursa-free vs. bursa-intact ovary. The difference in tumorigenic signals between them was non-significant (p=0.196 )

Reviewer 2 Report

The authors tried to establish a corresponding model to overcome the bursa barrier and demonstrate the metastasis-enhancing effect of ovulation in vivo.

This is a well-written and interesting paper.

However, in human, incessant ovulation is not the only cause of EOC. Incessant (retrograde) menstruation before menopause and bleeding caused by hormone replacement therapy after menopause are associated with the development of EOC including HGSC. (Hum Reprod 2011, Int J Mol Sci 2021)

I have one question. In this model, the effects of the volume of fluid may be associated with the development of dissemination. Is it possible that tumor cells in a higher volume of fluid (not FF) spread widely?

Author Response

(1) However, in human, incessant ovulation is not the only cause of EOC. Incessant (retrograde) menstruation before menopause and bleeding caused by hormone replacement therapy after menopause are associated with the development of EOC including HGSC. (Hum Reprod 2011, Int J Mol Sci 2021)

Response:

Agree with your point. In an earlier report, we showed hemoglobin, most likely sourced from retrograde menstruation, rescues the apoptosis of transforming FTE cells under ovulation-induced ROS stress (J Pathol. 2016 Dec;240(4):484-494). This may be part of the mechanisms of the pro-carcinogenic effect of retrograde menstruation.  

(2) I have one question. In this model, the effects of the volume of fluid may be associated with the development of dissemination. Is it possible that tumor cells in a higher volume of fluid (not FF) spread widely?

Response:

Indeed, as revealed in this study, superovulation lead to a significant higher tumor burden than the normal ovulation mice. Supposedly, superovulation would release a larger volume of follicular fluid.

Round 2

Reviewer 2 Report

Introduction. As the authors responded, retrograde menstruation appears to be involved in the HGS carcinogenesis in humans. Hence, incessant ovulation is not the cause of EOC, although it may be a major cause.

In the previous report, I commented on the possibility that non-FF (such as ascites associated with inflammation) may facilitate tumor cell dissemination. In this mouse model, a higher fluid volume in the peritoneal cavity was caused by superovulation. However, in humans, a higher fluid volume is observed in women with pelvic inflammation associated with retrograde menstruation and/or infection, and this type of fluid may help tumor cells to disseminate. The anatomical differences between mice and humans make extrapolating the results from mice to humans difficult.

The authors need to discuss the limitations of this study.

Author Response

RESPONSE TO REVIEWER (Round 2)

(1) Introduction. As the authors responded, retrograde menstruation appears to be involved in the HGS carcinogenesis in humans. Hence, incessant ovulation is not the cause of EOC, although it may be a major cause.

Agree with the point. This model did not take care of the physiological difference between the mouse estrus cycle (no menstruation) and the human menstrual cycle. We discussed this limitation in the enriched paragraph of discussion at the end of the article (see response 3). 

(2) In the previous report, I commented on the possibility that non-FF (such as ascites associated with inflammation) may facilitate tumor cell dissemination. In this mouse model, a higher fluid volume in the peritoneal cavity was caused by superovulation. However, in humans, a higher fluid volume is observed in women with pelvic inflammation associated with retrograde menstruation and/or infection, and this type of fluid may help tumor cells to disseminate. 

We agree with the point that pelvic inflammatory disease (PID), which is always associated with inflammatory peritoneal fluid, is associated with a higher risk of ovarian cancer [https://pubmed.ncbi.nlm.nih.gov/34320962/; https://pubmed.ncbi.nlm.nih.gov/32037193/ ]. An inflammatory microenvironment is known to contribute to tumor proliferation, angiogenesis, and metastasis [https://pubmed.ncbi.nlm.nih.gov/19770594/]. Meanwhile, ovulation is the consequence of acute inflammation after luteinization of the ovarian follicle, which releases rich ROS, inflammatory cytokines, and cells into the peritoneal cavity. We discussed this possible mechanism of metastasis promotion in the revised manuscript.

“Ovulatory FF harbors various molecules essential for folliculogenesis, ovulation, and postovulatory tissue regeneration [23]. Many of these molecules play essential roles in oncogenesis and metastasis. These include cytokines (IL-1β, IL-6) [24, 25], chemokines (SDF-1) [26, 27], growth factors (IGF2, HGF, TGF-β1, PDGF, FGF, and VEGF) [28-32], soluble ECM proteins (collagen I, III, IV, fibronectin, and vitronectin) [33-36], and adhesion molecules (ICAM-1and VCAM-1) [37-39]. Moreover, ovulation is a consequence of acute inflammation after luteinization of the ovarian follicle, which releases rich ROS, inflammatory cytokines, and cells to the peritoneal cavity. Pelvic inflammation is a known risk factor for ovarian cancer  [https://pubmed.ncbi.nlm.nih.gov/34320962/; https://pubmed.ncbi.nlm.nih.gov/32037193/]. Thus, ovulation may contribute to an inflammatory tumor microenvironment that promotes cell proliferation, angiogenesis, and cancer metastasis [https://pubmed.ncbi.nlm.nih.gov/19770594/].” 

(3) The anatomical differences between mice and humans make extrapolating the results from mice to humans difficult. The authors need to discuss the limitations of this study.

Thanks for the suggestion. We added more description in the paragraph discussing the limitation of the current model:

“The bursa-free mouse model, although essentially mimicking the ovulatory FF release in humans, is still limited by the fundamental anatomic and physiological differences between mice and humans. First, although we did not find a regeneration of the bursa up to xx weeks after the removal, whether the bursa-free state will affect ovarian physiology, such as the efficiency of ovulation and the postovulatory wound healing, is not known. The second concern is whether the direct drainage of FF into the peritoneal cavity will change the constitution of peritoneal fluid or affect the peritoneal environment. For this point, we do not think there will be a significant difference in the long run. Within the intact bursa, FF can drain to the peritoneum through the foramen of ovarian bursa, although at a slower speed [15]. Third, after mimicking bursa-free ovulation, a robust model simulating the menstruation and its backflow is still lacking, although there is a way to induce decidualization and blood-shedding in rodent endometrium by hormone and sesame oil treatment [https://pubmed.ncbi.nlm.nih.gov/33174022/]. “